# Appendiceal Mucinous Neoplasms: From Clinic to Pathology and Prognosis

**DOI:** 10.3390/cancers15133426

**Published:** 2023-06-30

**Authors:** Luis González Bayón, Lorena Martín Román, Pablo Lozano Lominchar

**Affiliations:** 1Peritoneal Carcinomatosis Unit, Department of General Surgery, Hospital General Universitario Gregorio Marañón, 28007 Madrid, Spain; pablo.lozano@salud.madrid.org; 2Faculty of Medicine, Universidad Complutense de Madrid, 28040 Madrid, Spain

**Keywords:** mucocele, pseudomyxoma peritoneal, mucinous appendiceal neoplasm, cytoreductive surgery, HIPEC, acellular mucina

## Abstract

**Simple Summary:**

The cecal appendix is known to the general population because it is where acute appendicitis develops and it usually needs a surgical intervention for treatment. However, it is also the place of origin of tumors with special behavior. Those tumors are uncommon and generate specific clinical situations, such as the mucocele or the pseudomyxoma peritonei syndrome, which are the subject of much debate regarding definitions and pathologic classification. There is great interest in achieving a common language for these tumors that will allow the sharing of research and treatment results which will improve the existing information and management of our patients.

**Abstract:**

Appendiceal mucinous neoplasms have been classified differently over time causing confusion when comparing results between working groups in this field and establishing a prognosis of the disease. A historical perspective of the different classification systems of these tumors is essential for the understanding of the evolution of concepts and histopathological definitions that have led up to the present moment. We carried out a systematic review of the pathological classifications of appendiceal mucinous tumors and how they have included the new criteria resulting from clinical and pathological research. The latest classifications by PSOGI and AJCC 8th edition Cancer Staging have made a great effort to incorporate the new pathological descriptions and develop prognostic groups. The introduction of these new classification systems has posed the challenge of verifying how they adapt to our casuistry and which one defines best the prognosis of our patients. We reclassified our series of patients treated for mucinous appendiceal tumors with cytoreductive surgery and hyperthermic intraperitoneal chemotherapy following the PSOGI and the AJCC 8th edition criteria and concluded that both classifications correspond well with the OS and DFS of these patients, with some advantage relative to the PSOGI classification due to a better histopathological description of the different groups.

## 1. Introduction

The cecal appendix is a remnant organ with an unknown function in humans. It does not exist in carnivorous or herbivorous animals but it does in omnivores. The mucosa of the appendix has a colonic-type columnar epithelium with neuroendocrine cells and mucin-producing goblet cells. The submucosa is rich in lymphatic tissue, which suggests that it may play a part in the immune system. Another functional hypothesis of the appendix is that it is a reservoir of good intestinal bacteria contributing to the maintenance of normal intestinal flora [1]. It is such that a relationship has been found between appendectomy and the development of colorectal carcinoma based on the alteration of the microbiome [2].

Acute appendicitis is the most frequent pathology that develops in the cecal appendix. However, the appendix is also the origin of tumors that display special behavior. The most frequent tumors that develop in the appendix are mucinous tumors. These tumors have been associated with specific clinical conditions such as mucocele or pseudomyxoma peritonei syndrome (PMP). However, both of these terms are subject to much debate since they have been widely used as pathological terms throughout the literature, resulting in confusion and overlapping terminology surrounding the pathological classification of these tumors. Currently, the term mucocele of the appendix can be used to describe the radiological or clinical finding of a dilated appendix. It results from the intraluminal accumulation of mucin secondary to either a mucinous tumor, or less commonly, an inflammatory obstruction. Mucoceles of the appendix represent a common way of presenting mucinous tumors of the appendix. Secondly, the term PMP is used to describe the clinical entity characterized by the accumulation of mucin within the peritoneal cavity. The most common origin of PMP has been established to be mucinous tumors of the appendix, however, PMP can also be the result of peritoneal dissemination (PD) from a mucinous tumor of the ovary or other organs in the abdominal cavity (pancreas, gallbladder, colon, urachus …). Therefore, the term PMP is no longer accepted as a pathological diagnosis.

The gold standard treatment for mucinous appendiceal neoplasms with PD is cytoreductive surgery plus hyperthermic intraperitoneal chemotherapy (CRS + HIPEC), which has substantially changed the prognosis of these patients but must be performed in specialized centers [3,4].

Despite optimal treatment with CRS + HIPEC, there is a wide range of clinical outcomes; some patients achieve excellent outcomes (i.e., high cure rates and low recurrence rates) while others show very aggressive behavior with early recurrences and limited survival. Prognosis is mainly determined by the histopathological features of the primary tumor and peritoneal deposits. However, determining prognosis has been complicated by the several classification systems available since the 1990s as a result of the constant evolution of pathological definitions. These have generated confusion amongst treating physicians and made the comparison of treatment results difficult. In addition, the correlation between prognosis and histology is not always accurate; this has encouraged the search for new indicators of biological aggressiveness.

In this article, we want to review clinical aspects and management guidelines of mucinous neoplasms of the appendix and PMP, and to expose the investigations carried out by our group on the mucinous neoplasm of the appendix:–A systematic review providing a historical perspective on the evolution of the different classification systems of these tumors published since the 1990s up to the 2016 Peritoneal Surface Oncology Group International (PSOGI) consensus, the 2017 AJCC 8th edition, and the 2019 WHO classification.–A pathological review of our series to the adoption of the PSOGI and AJCC 8th edition classification criteria in order to evaluate which classification system best reflects the prognosis of our cohort of patients.

## 2. Review Sections

### 2.1. Clinical Aspects

The conflict with mucinous lesions of the appendix starts in the clinic with the finding of an appendiceal mucocele. Rokitansky described, for the first time, the appendiceal mucocele [5], which is a clinical (currently radiological), surgical, or autopsy description of a dilated appendix by mucinous secretions retained due to an obstructed lumen. The cause of the obstruction is key. If secondary to chronic inflammation (fecalith, parasites, lymphoid hyperplasia, …) without signs of epithelial hyperplasia or neoplasia, it is a simple retention cyst (non-neoplastic mucinous appendiceal lesion) that can rupture and release mucin into the peritoneal cavity without developing PMP [6]. If, however, the obstruction is caused by an epithelial hyperplasia or neoplasia, with/without an associated epithelial degeneration process (adenomas, serrated lesions or mucinous neoplasm), then it is considered a neoplastic mucinous appendiceal lesion [7].

The other clinical debate surrounds PMP. PMP is a rare clinical entity characterized by the progressive accumulation of mucin throughout the peritoneal cavity, most likely resulting from a perforated mucinous neoplasm of the appendix [8]. Classically, it was diagnosed by the finding of a “jelly belly” at laparotomy (abdomen filled with mucinous ascites). The term PMP was initially introduced by Wert [9] in 1884 describing a case of mucinous ascites in the setting of a mucinous ovarian neoplasm. In 1901, Frankel described a similar case arising from a cyst or mucocele from the appendix [10]. PMP progresses in an indolent manner, being asymptomatic to mildly symptomatic with diffuse unspecific abdominal pain. Due to this, it is often found incidentally in patients undergoing radiological investigations or surgery for other reasons. Localized disease is frequently found in the setting of acute appendicitis, or investigations for right iliac fossa pain that may reveal a pelvic mass secondary to a dilated appendix or localized mucinous deposits. In cases of advanced disease, the classical clinical picture of a “jelly belly” can be observed. In these cases, patients have an increase in abdominal girth caused by the accumulation of mucinous ascites. Often, this is accompanied by the onset inguinal and umbilical hernias that appear as a consequence of increased intraabdominal pressure. Episodes of bowel obstruction may occur as a result of small bowel involvement representing the final stages of the disease. Disease progression leads to gradual abdominal distension and intestinal sub-occlusion, emaciation, and the impossibility of nutrition by mouth [11]. The surgical treatment used to be indicated late and generally consisted of an evacuation of mucin with the usual macroscopic tumor residue. Some patients had to undergo surgery every year to evacuate mucus, reaching a deadly situation of the impossibility of more mucinous evacuation due to visceral impregnation (solid tumors surrounding intestines, stomach, liver, spleen, bladder...etc.).

In 2000, Esquivel and Sugarbaker [12] evaluated the most common presentations of PMP in their series of 217 patients: acute appendicitis (27%), followed by abdominal distension (23%), vague abdominal pain (17%), or diagnosis of new onset hernia (14%).

A large population-based study from the Netherlands’ nationwide pathology database (Pathologic Anatomic National Automatic Archive, PALGA) found the incidence of PMP to be estimated at 2 per million inhabitants per year, with a certain predominance of women [13]. The incidence of appendiceal mucinous tumors (LAMN and HAMN) may be higher in interval appendectomy specimens from adults [14]. Alarming rates have been described in a recent randomized clinical trial performed in the Netherlands where an appendiceal mucinous neoplasm was found in 20% of cases above the age of 40 with a previous periappendiceal abscess [15]. A meta-analysis found a pooled prevalence of appendiceal neoplasms at interval appendectomy after an episode of complicated appendicitis of 11% (95% CI 7–15%) [16]. In this setting, the most frequent type of appendiceal primary tumor found was mucinous neoplasm (43%). A recent population-based study from the SEER database identified that the risk of appendiceal adenocarcinoma or PMP was significantly higher in patients with a periappendiceal abscess (OR 15.05, *p* < 0.0001, perforated appendicitis (OR 4.09, *p* = 0.0018), and patients above the age of 40 (OR 26.46, *p* < 0.0001) [17].

### 2.2. Endoscopy and Imaging Modalities

Many patients end up having an endoscopic evaluation of the digestive track as part of the investigations carried out for vague abdominal symptoms. Endoscopies are inevitably normal though the presence of seeping mucin through the appendix orifice can be pathognomonic but infrequently seen. In the setting of PMP or an incidental finding of a mucinous neoplasm, the European and American guidelines recommend an endoscopic evaluation in order to rule out the presence of synchronous colorectal neoplasia [3,4]. A German multicenter study described rates of synchronous colorectal neoplasia in 8.9% of patients with a mucinous neoplasm of the appendix [18].

The most commonly used diagnostic modality for PMP is the computed tomography scan (CT) of the chest, abdomen, and pelvis with an intra-venous and oral contrast. The PMP expert panel agreed with a 96,4% consensus that CT imaging is the preferred preoperative imaging modality [3]. The benefits of CT imaging include its accessibility, low cost, and easier interpretation by radiologists not experienced in peritoneal malignancies.

The classical radiologic features of PMP include omental caking, mucinous ascites, and scalloping of the liver [19]. These features, however, are only present in the setting of an advanced disease. CT imaging can also reveal a mucocele of the appendix, which refers to the radiological image of a mucin-filled and distended appendix that may be accompanied by peripheral calcification. In the setting of acute appendicitis, a retrospective study including 65 patients reported a sensitivity of 95% in the detection of appendiceal tumors using CT imaging when taking morphologic criteria (i.e., cystic dilation or presence of a soft tissue mass) or an appendiceal diameter greater than 15 mm [20].

On the other hand, the detection of peritoneal disease represents an ongoing challenge with current imaging techniques. The detection of peritoneal deposits is determined by their size and location. The sensitivity of CT to detect lesions greater than 5 cm ranges from 59–94% but drops to 19–28% in lesions smaller than 1 cm and to 11–28% in lesions smaller than 0.5 cm [3,21]. Sensitivity also depends on the involved area. To elaborate, one retrospective study found that lesions in the ileocecal area had the lowest sensitivity for detection (11–28%), followed by lesions in the right subdiaphragmatic area (11–22%) and the omentum and transverse colon (25%). The detection rate of lesions in the small bowel and/or its mesentery ranges from 18 to 55% [22].

Recently, the use of diffusion-weighted magnetic resonance imaging (DW-MRI) has been shown to improve both sensitivity and specificity in the detection of peritoneal metastasis with a sensitivity of 85–90% in cases of peritoneal deposits smaller than 1 cm [23]. A study published by Low et al. comparing preoperative MRI with CT reported that MRI predicted tumor volume accurately in 91% of the patients and CT only in 50% [24]. Additionally, MRI was able to detect diseases in the small bowel in 92% of the cases whereas CT did so in only 48%. These findings were backed up by the results from a larger study that also highlighted that MRI requires experienced radiologists for its accurate interpretation, especially when evaluating the small bowel [25].

The presence of disease in certain locations is associated with a worse prognosis due to the reduced likelihood of achieving optimal CRS. These locations are the small bowel and/or its mesentery, the porta hepatis and hepatoduodenal ligament, ureteric encasement, and biliary obstruction [26]. Such observations led to the study and development of radiologic scores to predict the resectability of the disease. One example of these scores is the simplified preoperative assessment for appendix tumors (SPAAT) score [27]. This score was developed to predict the ability of complete CRS in patients with low-grade peritoneal disease originating from the appendix based on the following findings on preoperative CT imaging: scalloping of the liver, pancreas, spleen, or portal vein (1 point each) and the presence or absence of mesenteric foreshortening of the small bowel (from 0 to 3 points). The score was externally validated on a larger cohort with a score of <3 accurately predicting a complete CRS in 97.1% of the cases. Another example is the simplified radiographic score (SRS) [28]. This score takes into consideration the thickness of the disease (measured in millimeters) in five regions of the upper abdomen (inferior vena cava to the portal vein, right hepatic lobe to left hepatic lobe, left hepatic lobe to lesser sac, Spiegel lobe to left hepatic lobe, and Spiegel lobe to right crus). A sum greater than 28 mm predicts incomplete CRS with a positive predictive value of 85% and a negative predictive value of 59% in their validation cohort. The discriminative capacity of these scores has been further investigated by other study groups. The usefulness of the SPAAT score was questioned after observing a positive predictive value of 50% with a sensitivity of 40%, even though the cut-off of three was associated with optimal CRS in the binary regression model [21]. A second study group aimed to evaluate the utility of both scores and observed positive predictive values of the SPAAT and SRS scores of 67% and 75%, respectively [29].

These scoring systems, however, have not been widely adopted and there still exists a lot of variability in the reporting of radiologic imaging of patients with peritoneal disease. With the aim of standardizing reporting, the Peritoneal Malignancy Institute in Basingstoke proposed the PAUSE method [30]. This acronym stands for P: PCI score; A: abdominal wall and ascites; U: unfavorable sites of disease (periportal, the root of mesentery, ligament of Treitz, pelvic side wall disease, and disease involving the sacrum); S: small bowel and mesenteric disease; and E: extraperitoneal disease. Following this method, the radiology report will contain information that will aid in the selection of patients who will benefit from CRS + HIPEC at the MDT as well as facilitating research across different centers.

### 2.3. Tumor Markers

The prognostic implication of tumor markers in PMP has been broadly studied by different study groups over the course of time. The tumor markers associated with PMP are the carcinoembryonic antigen (CEA), cancer antigen 19-9 (Ca19-9), and cancer antigen 125 (Ca-125). The results obtained have not been homogeneous across the different study groups. However, current guidelines [3,4] recommend tumor markers to be included as part of the preoperative work-up for these patients.

The elevation of all three tumor markers has been associated with a higher volume of disease [31] and with a reduced probability of achieving optimal CRS [32,33]. The study group from the peritoneal malignancy unit in Milan concluded that preoperative CEA, Ca19-9, and Ca-125 could be used as predictors of optimal CRS [33] and reported that Ca19-9 and Ca-125 were more powerful predictors of prognosis than histology. Similar findings were observed by the study group in Sydney [34]. The elevation of tumor markers preoperatively has also been associated with shorter OS and DFS outcomes [32,35]. CEA elevation has been associated with shorter DFS [36], whereas other study groups found Ca19-9 to be a predictor of shorter DFS [34,37].

### 2.4. Preoperatory Hystology

Increased awareness of appendiceal PMP and developments in cross-sectional imaging has increased the detection capacity of appendiceal lesions. Therefore, it is currently under debate whether histological confirmation is necessary in the setting of typical radiological findings [3,4,38]. In case of diagnostic doubt, histological confirmation is recommended, preferably by means of diagnostic laparoscopy or core needle biopsy. Fine needle aspiration usually fails to sample representative tissue and frequently results in acellular mucin.

Exploratory laparoscopy should be carried out in a referral center and by placing the ports in a midline position [38]. The advantages of laparoscopy include the possibility of taking a proper biopsy under direct vision, assessment of small bowel and mesenteric involvement, and an estimation of the PCI score. European guidelines do highlight that staging with a proper evaluation of the small bowel and its mesentery plays an important role in patients with a high-grade disease where initial treatment with systemic chemotherapy (SCT) might be indicated [3].

### 2.5. Peritoneal Dissemination

Primary mucinous appendiceal neoplasms have a tendency to disseminate to the peritoneum. PMP is characterized by the “redistribution phenomenon”, a term introduced by Sugarbaker to explain the accumulation of mucin at predetermined anatomical sites and its absence at others [39]. The most widespread model of how PMP develops is through the rupture of an appendiceal mucinous neoplasm allowing mucinous material with or without neoplastic cells to access the peritoneal cavity (transcoelomic spread). Consequently, the mucin and cells enter the peritoneal fluid circulation in a clockwise direction as dictated by the peristaltic movement of the gastrointestinal tract, the effects of gravity, and the suction effect of the movement of the diaphragms [40]. Mucus accumulation occurs in sloping areas of the abdominal cavity (Douglas pouch) and in areas of intraperitoneal fluid reabsorption (diaphragms and greater omentum).

Tumor implantation is the next step of the peritoneal metastatic cascade. For this, tumor cells have to reach the submesothelial space to develop a tumor growth colony (peritoneal implant). A total of two mechanisms have been described for this: the transmesothelial route and the translymphatic route. In the transmesothelial route, free cancer cells attach to the mesothelium through adhesion molecules (CD44, integrins, selectins, etc.), liberating cytokines (interleukins, EGF, HGF, …) that induce the contraction of mesothelial cells exposing the submesothelial basement membrane [41,42]. This allows tumor cells to adhere to the submesothelial plane through integrins and promote stromal invasion by releasing metalloproteinases [43]. Jayne et al. suggested that tumor cells provoke the apoptosis of mesothelial cells, which they confirmed with DNA fragmentation [44].

On the other hand, in the translymphatic route, free cancer cells gain access to the subperitoneal lymphatic space through lymphatic stomata. Lymphatic stomata are openings between mesothelial cells that are communicated with lymphatic capillaries where peritoneal fluid reabsorption takes place. These areas contain many macula cribiformis in the underlying connective tissue with a lot of oval foramina where the submesothelial space is exposed [45]. Associated with lymphatic stomata are other structures named milky spots which consist of macrophages, lymphocytes, and plasmatic cells supported by blood and lymphatic vessels that surround stomata in specific places such as greater omentum, appendices epiploic of the colon, diaphragm, falciform ligament, Douglas pouch, and the interface of small bowel mesentery with the intestinal tube. These structures, lymphatic stomata, macula cribiformis, and milky spots seem to play a role in the translymphatic route that seems to be the main way of peritoneal implantation in PMP [46,47]. The other main conditioning factor of the redistribution phenomenon is gravity, explaining why tumor aggregates are commonly found in the pouch of Douglas, the paracolic gutters, and the retrohepatic space [48].

### 2.6. CRS + HIPEC

In the past, peritoneal carcinomatosis was considered to be an incurable condition identical to that of distant metastases. Lack of response to the systemic treatments available at the time led to the development of aggressive locoregional treatments based on the hypothesis that peritoneal carcinomatosis is a loco-regional disease. In 1979, the first CRS + HIPEC procedure was performed by John Spratt on a PMP patient [49] at the University of Louisville, Missouri. Spratt was working on dog models of peritoneal carcinomatosis and had designed a hyperthermic chemotherapy peritoneal perfusion machine. The patient was a 35-year-old man who had a 2-year history of abdominal distension and had been diagnosed with PMP during a laparotomy. The histopathological study showed a mucin-producing low-grade adenocarcinoma of unknown origin. The patient asked Spratt to be the first case to undergo CRS + HIPEC. The procedure performed was an extensive cytoreduction followed by the intraperitoneal perfusion with 105 mg of thiotepa at 42 °C over 15 min with the abdomen closed. The intraperitoneal perfusion procedure was repeated 5 days later with 75 mg methotrexate over 30 min. The postoperative period was uneventful and the pathologist finally identified the PMP to be originating from the pancreas. No subsequent follow-up is known.

The combination of CRS + HIPEC has been established to be the standard of care for PMP patients as per the recently published European [3] and American treatment guidelines [4]. Given the low incidence of PMP, randomized controlled trials evaluating the benefit of CRS + HIPEC in this setting are missing. Therefore, evidence supporting the use of CRS + HIPEC is mainly based on results from retrospective series. Concerns regarding the use of CRS + HIPEC were raised after the results of the PRODIGE-7 trial were published, where adding HIPEC with oxaliplatin to CRS of peritoneal metastases from colorectal cancer did not provide any survival advantage but did increase postoperative morbidity [50]. This triggered the need to investigate the efficacy of HIPEC treatment in PMP patients. A large multicenter cohort study from the PSOGI registry included patients with PMP treated with CRS alone (*n* = 376) or CRS + HIPEC (*n* = 1548) and concluded that HIPEC was associated with better survival outcomes (hazard ratio (HR) 0.65 (95% CI 0.50–0.83), *p* = 0.001) without an increased risk of morbidity throughout the entire series (HR 0.94 (95% CI 0.65–1.37), *p* = 0.76) [51].

Nonetheless, CRS + HIPEC is a complex procedure with associated morbidity and mortality rates that cannot be overlooked. Performance status and the frailty of patients have to be evaluated [52]. Recent prospective randomized trials have estimated severe morbidity and mortality rates to stand between 25–27% and 0–2%, respectively [50,53]. In the setting of PMP, a recently published meta-analysis including 13 studies reported a major complication rate (Clavien-Dindo ≥ 3) for 1747 patients of 32.9% (95% CI 30.5 to 35.4%) [54]. Therefore, careful patient selection is vital. The decision to proceed with CRS + HIPEC must take place at a multidisciplinary meeting with an expert radiologist, pathologist, surgical oncologist, and medical oncologists in order to aid individualized treatment strategies.

### 2.7. Systemic Chemotherapy (SCT)

The role of SCT in the treatment of patients with PMP is unknown. There exists an overall lack of evidence regarding SCT treatment in terms of selecting which patients would benefit from it and which regime. This task is made difficult by the overlapping and confusing terminology that surrounded this pathological entity in the past, and its low incidence that obstructs the possibility of conducting randomized clinical trials. Additionally, experts recommend the use of the same treatment regimens approved for colorectal cancer [55], even though the natural course, biological, and molecular profile are different [56,57].

Of note, one important contribution is the result of a large retrospective analysis using the National Cancer Database (NCBD) including 5971 mucinous tumors of the appendix [58]. Their relevant observation was that SCT benefited stage IV patients with moderately (median OS 2.99 vs. 1.64 years, *p* = 0.0005) and poorly differentiated diseases (1.57 vs. 1.02, *p* = 0.0007) but did not influence those with well-differentiated diseases [58]. Reinforcing this, a later publication using the NCDB including only patients with metastatic low-grade mucinous appendiceal adenocarcinoma showed no survival advantage when SCT was associated with the treatment of these patients [59]. These results have dictated recommendations in treatment guidelines. Both the European [3] and North American guidelines [4] do not recommend the use of SCT in low-grade diseases but contemplated the use of SCT in high-grade diseases in the neoadjuvant and adjuvant settings, although, it is specified that the level of evidence to support these recommendations is low. The guidelines coincide in favoring neoadjuvant SCT in cases with initially unresectable diseases.

There have been few studies investigating whether neoadjuvant SCT provided any survival benefit in the treatment of PMP with some disparity among the results. Optimistic results were reported by Bijelic et al. [60]. The rate of histologic responses observed in this retrospective series of high-grade PMP patients was 29%; this was associated with improved OS (*p* = 0.003). Neoadjuvant SCT was also associated with lower PCIs (19 vs. 28, *p* = 0.003). Additionally, Spiliotis et al. [61] published higher OS and DFS in high-grade PMP patients following neoadjuvant SCT and CRS + HIPEC (median OS 19 vs. 10 months, *p* = 0.042 and median DFS 10 vs. 0 months, *p* = 0.039). However, a similar study by Turner et al. [62] reported a radiological response rate of 58% that did not translate into significant changes in the PCI, CC score, or survival outcomes. Lieu et al. [63] analyzed the subgroup of patients with poorly-differentiated PMP with SRC and observed improved DFS rates with SCT. Milovanov et al. [64] described a 3-year OS of 22% with SCT vs. 14% without SCT; *p* = 0.028.

On the other hand, the study by Votanopoulos et al. [65] found preoperative SCT treatment to be a factor in a worse prognosis for both low-and high-grade PMP patients. Recently, a report from the Aerospace Center Hospital, Beijing, China, on 750 patients with PMP treated with CRS + HIPEC also concluded that preoperative SCT was associated with worse survival outcomes in low-grade PMP patients, PCI < 20, and optimal CRS. In addition, it did not provide any survival advantage in patients with high-grade PMP, PCI > 20, or in cases of suboptimal CRS [66].

The results from studies evaluating the effect of adjuvant SCT are similarly inconclusive. Blackham et al. [67] investigated the effect of perioperative SCT in the treatment of low- and high-grade PMP patients. The results were that the SCT response rate observed in low-grade PMP was 0%. In cases of high-grade PMP, the DFS was higher in patients receiving adjuvant SCT (13.6 vs. 7.0 months, *p* = 0.03) but did not translate into an improvement in OS (36.4 vs. 19.4 months, *p* = 0.14). Therefore, they concluded that high-grade PMP patients could benefit from the use of adjuvant SCT and would recommend using SCT in the neoadjuvant setting only in borderline resectable cases. These same recommendations can be seen in the European and North American guidelines [4].

### 2.8. Pathology

The development of classification systems for PMP based on histopathological findings and prognosis has been challenged by the existence of confusing and overlapping terminology.

In 1995, Ronnet et al. [68] published their pivotal study where three different pathological subgroups with their respective prognoses were defined. Disseminated peritoneal adenomucinosis (DPAM) was used to refer to peritoneal lesions formed by abundant extracellular mucin and scant epithelial cells with little atypia or mitotic activity. Peritoneal mucinous carcinomatosis (PMCA), on the other hand, was used to describe peritoneal lesions composed of abundant mucinous epithelium, with high mitotic activity and cytological features of carcinoma. An intermediate category grouped lesions with discordant characteristics (PMCA-I/D) with features of DPAM and PMCA. Patients with DPAM had a significantly better prognosis than those with PMCA-I/D or PMCA (age-adjusted 5-year OS rates were 84% versus 37.6% and 6.7% respectively; *p* < 0.0001).

In 2003, Misdraji et al. [69] introduced the term LAMN to describe primary mucinous lesions of the appendix with a pushing invasion that could perforate the wall of the appendix and disseminate through the peritoneal cavity, unlike the adenocarcinoma (MACAs) that presented an infiltrative invasion. LAMNs showed low-grade cytologic atypia (rare mitotic figures, nucleomegaly, and scarce nuclear stratification) and minor architecture complexity (flat epithelial proliferation and small papillary excrescences). Conversely, mucinous adenocarcinomas of the appendix (MACA) were depicted by the infiltrative invasion of the appendiceal wall with high cytologic atypia (full-thickness nuclear stratification, vesicular nuclei, prominent nucleoli, nuclear membrane irregularities, and brisk mitotic activity). When there was PD, they defined a two-tiered system: LAMN involving the peritoneum with a better prognosis and MACA involving the peritoneum (5-year OS 86 vs. 44%; *p* = 0.04). The presence of SRCs was an exclusion criterion from the study.

Subsequently, Bradley and colleagues [70] reviewed the histology of 101 cases of PMP to assess the prognostic implications of Ronnet’s three-tiered classification. DPAMs, previously attributed to adenomas in Ronnett’s classification, were credited with primary LAMNs whereas PMCAs (high-grade atypia and/or SRC) were associated with moderate or poorly differentiated appendiceal adenocarcinomas. They did not find differences between the survival outcomes of the DPAM and PMCA-I groups. However, the PMCA group did have a significantly worse 5-year OS. Therefore, they proposed a two-tiered classification system whereby SRCs were included in the PMCA subgroup. The terminology they advocated for was low-grade mucinous carcinoma peritonei (MCP-L) and high-grade mucinous carcinoma peritonei (MCP-H).

Primary appendiceal tumors were classified into LAMN, MACA, SRC carcinoma, and undifferentiated appendiceal carcinoma in the WHO 4th edition 2010 [71]. The two-level of peritoneal lesions were defined as low-grade and high-grade diseases. Low-grade disease entailed acellular content or few cells forming islands or strands, with light cytologic and nuclear atypia, and scanty mitoses. The high-grade disease was characterized by a greater number of cells, arranged in strands or islands, high-grade atypia, and frequent mitosis. The existence of SRCs led directly to a high-grade lesion. However, at this time, PMP was still recognized by the WHO as a pathological diagnosis and a borderline malignant entity. Carr and co-workers [72] validated the prognostic power of the two-level staging system of the WHO 4th edition classification and found significant differences in OS between low-grade and high-grade PMP treated with CRS + HIPEC (5-year OS 84 and 48%; *p* < 0.001).

Additionally, in 2010, the AJCC 7th edition Cancer Staging Manual [73] separated appendiceal carcinomas from colorectal carcinomas for the first time and distinguished between mucinous and non-mucinous adenocarcinoma subtypes. For primary lesions, they introduced a classification system similar to that of colorectal cancer: well-differentiated (G1), moderately differentiated (G2), and poorly differentiated (G3) tumors. Histological grade was included in the stage IV disease but only two histological prognostic groups were recognized: stage IVA (G1, N0) low-grade mucinous adenocarcinomas and stage IVB (G2-3, and any G with N positive) high-grade mucinous adenocarcinomas. Milovanov et al. [74] endorsed the new AJCC 7th-edition staging classification, however, they defined two prognostic groups into stage IVA (DPAM constituted a specific good prognostic group). On the other hand, in a large retrospective database study by Overman et al. [75], the outcomes for stage IVB were different between moderately differentiated and poorly differentiated mucinous adenocarcinoma; hazard ratios (HRs) were 1.63 (95% CI: 1.14–2.34) and 4.94 (95%CI: 3.32–7.35), respectively. Thus, the combination of moderate and poorly differentiated diseases in the same prognostic group was not supported either.

Davison et al. [76] identified some pathological features of PMP that were correlated with worse survival outcomes such as destructive invasion, high cytologic grade, high tumor cellularity, angiolymphatic invasion, perineural invasion, and the presence of signet ring cells (SRC). They gave a better pathologic description to the three-tiered classification of the AJCC 7th edition staging classification: grade G1 included cases without adverse features; G2, included at least one adverse feature except SRC; and G3 were those with the presence of SRC. They found that G2 and G3 had a significantly worse prognosis than G1 (*p* < 0.0001). In the multivariate analysis, G2 presented a HR 2.7 (95% CI, 1.2–6.2) and G3 5.1 (95% CI, 1.7–14) relative to G1.

Significant progress in the pathological reporting of appendiceal mucinous neoplasms and PMP took place after the PSOGI meeting in 2016. In this meeting, experts in PMP from around the world voted on pathological terminology and its corresponding descriptions following a Delphi process [77]. The Group supported the terms LAMN and high-grade appendicular neoplasm (HAMN) for primary tumors with low or high cellular atypia, loss of lamina propria and muscularis mucosae, fibrosis of submucosa, and pushing invasion of the wall by acellular mucin or mucin with epithelial cells. The term cystadenoma was discarded for the appendix, and the term adenoma was chosen to refer to those colorectal type lesions confined to the mucosa and with intact muscularis mucosae, and the term serrated polyp to describe a lesion with serrated features and intact muscularis mucosae. They emphasized differentiating between pushing-invasion shown by LAMNs and HAMNs, and infiltrative invasion which characterizes the adenocarcinoma. Mucinous adenocarcinomas showed destructive invasion with tumor budding and/or small, irregular glands within a desmoplastic stroma and they were classified into well, moderately, or poorly differentiated types. The presence of SRC directly leads to the poorly differentiated type and implies an aggressive disease with poor clinical results. The two types of primary tumors with SRC were defined: mucinous adenocarcinoma with SRCs if less than 50 percent of the tumor cells were SRCs and mucinous SRC carcinoma when SRCs account for more than 50 percent of the tumor cells.

In the scenery of peritoneal dissemination, the grade of the peritoneal disease defined the prognosis and four prognostic groups were identified: acellular mucin (AM), low-grade mucinous carcinoma peritonei (LGMCP), high-grade mucinous carcinoma peritonei (HGMCP), and high-grade mucinous carcinoma peritonei with signed ring cells (HGMCP-SRC). AM is at the less aggressive extreme of the scale, whereas HGMCP-SRC is the most aggressive. The remaining two intermediate categories are LGMCP (peritoneal implants with <20% cellularity, little atypia, and few mitoses) and HGMCP (peritoneal implants with >20% cellularity, marked proliferative activity, and atypia but without SRC). At least >10% of the SRC component was required for a patient to be classified as HGMCP-SRC (Table 1). Groups with epithelial cells are comparable to those established by Davison et al. [76] G1, G2, and G3 and Shetty and colleagues [78] PMP 1, PMP 2, and PMP 3. Baratti et al. [79] reclassified 265 PMP patients treated with CRS + HIPEC following the criteria of the PSOGI classification but failed to validate its four prognostic groups. Instead, the two-tiered classification of the WHO 4th edition was supported.

The AJCC 8th edition Cancer Staging Manual 2017 incorporated the advances in terminology from the PSOGI consensus [80]. A great effort has been done to adapt TNM to the definition of LAMN in the T category. Tis (LAMN) referred to low-grade mucinous neoplasia with at least the loss of the muscularis mucosae; however, it could extend to the submucosa and muscularis propria by pushing invasion without changing the prognosis, making T1 and T2 categories not applicable. LAMN pT3 involves the subserosa and LAMN pT4 implies involvement of the serosa as with other carcinomas. HAMNs were classified using the same staging system as adenocarcinomas since they present a more aggressive clinical course. Moreover, stage IV disease was classified by M and G categories. The M category was divided into M1a, intraperitoneal dissemination of acellular mucin; M1b, peritoneal implants with tumor cells; and M1c, non-peritoneal metastasis. The G category was divided into three groups based on cytological features, tumor cellularity, and the presence of SRCs. G1 presented low-grade cytological atypia, <20% cellularity, without SRCs, and agreed to a well-differentiated adenocarcinoma. G2 corresponded to a moderately differentiated mucinous adenocarcinoma with high cytological atypia, and >20% cellularity without SRCs. Finally, G3 stated a poorly differentiated adenocarcinoma defined by any component of SRCs. Finally, the peritoneal disease was staged into two prognostic groups: stage IVA was defined by M1a (acellular mucin) or M1b G1 (low-grade atypia); IVB by M1b G2 (high-grade atypia) or G3 (high-grade atypia with SRCs). Stage IVC was M1c (non-peritoneum distant metastases).

The 2019 WHO 5th edition [81] classified appendiceal epithelial lesions into LAMN, HAMN, and adenocarcinomas (mucinous adenocarcinoma, mucinous adenocarcinoma with SRC, SRC carcinoma, and nonmucinous adenocarcinoma colorectal type). Goblet cell adenocarcinomas and neuroendocrine neoplasias were also included within the epithelial lesions, as well as two types of benign tumors: hyperplastic polyps and sessile serrated lesions (with sparing of the muscularis mucosae) [82]. Peritoneal dissemination was classified into G1 for low-grade peritoneal mucinous neoplasia; G2 for high-grade peritoneal mucinous neoplasia, and G3 for high-grade peritoneal mucinous neoplasia with SRC.

### 2.9. Other Histopathological Landmarks

#### 2.9.1. Acellular Mucin

Pai and colleagues [83] observed that only one patient out of 14 with AM recurred after 45 months. The existence of acellular or cellular mucin was associated with OS in the multivariable analysis. Furthermore, Davison and co-workers [76] described that 7% of patients with LAMN presented AM deposits and none of them recurred. Therefore, based on these outcomes, patients with peritoneal dissemination of acellular mucin present a lower risk of relapse with subsequently higher OS outcomes than those with low-grade cellular diseases.

#### 2.9.2. Signet Ring Cells

The presence of SRCs has also been a matter of much discussion. In 2014, Sirintrapum and co-workers [84] studied the significance of SRCs in 55 patients with MACA and PD. None of the 11 patients with low-grade adenocarcinoma had SRCs, whereas 29 of the 44 in the high-grade adenocarcinoma group presented SRCs. There were two types of SRC described: SRCs floating in mucin pools or tissue-invading SRCs. The 5-year OS for patients with MACA with SRCs in mucin pools was similar to that of patients with high-grade mucinous adenocarcinoma without SRCs (36% versus 32%, respectively; *p* = 0.58). The presence of SRCs invading tissues decreased OS to a median of 0.5 years, compared to 2.9 and 2.4 years for high-grade mucinous adenocarcinoma without SRCs (*p* = 0.003) and mucinous adenocarcinoma with floating SRCs (*p* = 0.004). Moreover, mucinous adenocarcinoma with SRCs invading tissues was associated with a higher rate of incomplete cytoreductions.

### 2.10. Prognostic Factors

As previously highlighted, the correlation between histology and prognosis has been widely studied throughout the literature. However, several other prognostic factors have been identified.

It is not surprising that the PCI and CC scores have been repeatedly associated with survival outcomes by several study groups. At least four study groups found the PCI score to correlate with OS in the multivariable analysis with cut-offs of PCI > 20 [85,86,87] and PCI > 22 [79] to be associated with worse OS. Similarly, the CC score of 2–3 was associated with worse survival outcomes by several study groups [79,85,86,88,89,90]. LN status is another factor that has been associated with OS in several studies [85,86,90]. Overall, these factors should be taken into consideration when selecting patient candidates for CRS + HIPEC. Solomon et al. [87] in their study on SRC cases with PD from the appendiceal, colorectal, and gastric origin, argued that high PCI does not contraindicate CRS + HIPEC but MDT discussion is warranted in order to evaluate whether optimal CRS is achievable. The study group of Levinsky et al. [90] came to a similar conclusion; CRS + HIPEC can be considered in SRC patients given the absence of LN metastasis and if CC0/1 can be achieved. However, the PCI, CC score, and LN status are factors that are either determined intra- or postoperatively. Prognostic factors that are determined preoperatively are needed in order to aid in the patient selection process [85].

Other factors that were found to be associated with prognosis by isolated study groups were severe postoperative complications [87,91] preoperative SCT [79], elevated Ca19-9 [92], and intraoperative transfusion [92].

**Table 1 cancers-15-03426-t001:** Main histologic classification systems [93]. SRC—signet ring cells; DPAM—disseminated peritoneal adenomucinosis; PMCA—peritoneal mucinous carcinomatosis; PMCA-I/D—peritoneal mucinous carcinomatosis with intermediate/disconcordant features; LAMN—low-grade appendiceal mucinous neoplasm; MACA—mucinous adenocarcinoma of the appendix, HAMN—high-grade appendiceal mucinous neoplasms; LG-MCP—low-grade mucinous carcinomatosis peritonei; HG-MCP—high-grade mucinous carcinomatosis peritonei.

	Stage of Disease	Type	Histological Nomenclature	Key Histologic Features
Ronnett et al. [68]	Primary tumors	Benign lesions.	Villous adenoma	Adenomatous epithelium with villous architecture confined to the mucosa.
			Cystadenoma	Adenomatous epithelium without villous architecture confined to the mucosa of a dilated appendix.
			Dilated/ruptured adenoma.	Glands or strips of adenomatous epithelium within the wall or on the serosa of a dilated or ruptured appendix without a stromal response. Dissecting mucin or epithelium extending through the wall of the appendix.
		Invasive lesions	Adenocarcinoma	Adenomatous epithelium invading the muscularis of the appendix accompanied by a stromal response.
			Mucinous adenocarcinoma with SRC	Neoplasms with glandular and SRC differentiation, with or without neuroendocrine features that showed marked cytologic atypia and muscularis invasion.
	Peritoneal implants		DPAM	Scant strips of simple proliferative epithelium with minimal to moderate cytologic atypia and no significant mitotic activity within abundant mucin.
			PMCA I/D	Features of DPAM with focal areas of carcinoma +/− SRC. I- Arising from a well-differentiated mucinous adenocarcinoma. D- Arising from a villous adenoma with moderate to marked cytologic atypia and areas of poorly differentiated carcinoma in the wall and serosa of the appendix.
			PMCA	Abundant proliferative epithelium, glands, nests, or individual cells including SRC, demonstrating marked cytologic atypia and mitotic activity.
Misdraji et al. [69]	Primary mucinous tumors		LAMN	Low-grade cytological atypia (nuclear enlargement, scarce nuclear stratification, and rare mitotic figures) and minimal architectural complexity (a uniform, flat, epithelial proliferation forming small papillary excrescences). No infiltrative invasion of the appendiceal wall.
			MACA	High cytological atypia (full thickness nuclear stratification, vesicular nuclei with prominent nucleoli and brisk mitotic figures) and infiltrative invasion of the appendicular wall.
	Peritoneal implants.		LAMN with peritoneal dissemination.	Low-grade cytologic atypia with flat epithelium proliferation forming papillary excrescences, low cellularity.
			MACA with peritoneal dissemination.	High-grade cytologic atypia, destructive invasion of the wall of the appendix, high cellularity, and abundant mitotic figures.
PSOGI classification [77]	Primary mucinous tumors.	Benign lesions.	Serrated polyp with or without dysplasia.	Tubular architecture with basal parts of the crypts showing serration, and dilatation. Muscularis mucosae intact.
		Mucinous neoplasms.	LAMN	Pushing invasion with loss of the muscularis mucosae and fibrosis of the submucosa. Filiform villi, undulating and flat. Basally orientated nuclei with minimal atypia and rare mitotic figures.
			HAMN	Pushing invasion with loss of the muscularis mucosae. Filiform villi, undulating, flat with pseudopapillae. Loss of nuclear polarity and frequent mitotic figures that may be atypical.
			Mucinous adenocarcinoma	Infiltrating invasion (discohesive single cells or clusters of cells, small irregular glands within desmoplastic stroma). Variably sized glands and islands, and variable nuclear features and frequent mitotic figures that may be atypical. Can be well-, moderately-, and poorly differentiated.
			Mucinous adenocarcinoma with SRC.	Infiltrating invasion. Poorly differentiated, with <50% of SRC.
			SRC carcinoma.	Infiltrating invasion. Poorly differentiated, with >50% of SRC.
	Peritoneal implants	No epithelial component.	Mucin without epithelial cells.	Acellular mucin. Abundant mucin without evidence of neoplastic epithelium. Extensive sampling required to discard presence of neoplastic epithelium.
		Epithelial component	LG-MCP	Abundant mucin with low cellularity (<20% tumor volume composed of neoplastic epithelium). Low-grade cytological features with low proliferative activity.
			HG-MCP	Abundant cellularity (>20% tumor volume composed of neoplastic epithelium). High-grade cytological features with high proliferative activity (can be mixed with areas of low-grade cytological features). Infiltrative invasion into subjacent tissues. Must lack SRC.
			HG-MCP with SRC	Abundant cellularity (>20% tumor volume composed of neoplastic epithelium). High-grade cytological features with high proliferative activity. Infiltrative invasion into subjacent tissues. SRC component present.
AJCC 8th edition [80]	Primary lesions.	Benign lesions	Adenoma	LAMN confined to the mucosa with intact muscularis mucosae.
		Premalignant lesions	High-grade dysplasia	Neoplastic cells confined to crypts that do not invade the lamina propria.
			Intramucosal adenocarcinoma	Neoplastic cells invade the lamina propria with or without extension into but not through the muscularis mucosae. pTis.
		Mucinous appendiceal neoplasms	LAMN	Neoplastic cells extend through the wall of the appendix with a pushing front, without features of invasion. Tis (LAMN)- LAMN confined by the muscularis propria, acellular mucin, or mucinous epithelium may extend into de muscularis propria. pT3- involvement of the subserosa. pT4a- involvement of the visceral peritoneum (with acellular mucin or mucinous epithelium). pT4b- direct involvement of adjacent organs or structures.
			HAMN	Tumors with architectural features of LAMN with areas of high-grade dysplasia. pT staging follows that of mucinous adenocarcinoma.
			Mucinous adenocarcinoma.	Neoplastic epithelium displays infiltrative and destructive growth into the wall of the appendix, beyond the muscularis mucosae. Associated with desmoplastic reaction. pT1- involvement of the submucosa through the muscularis mucosa. pT2- involvement of the muscularis propria. pT3- involvement of the subserosa or mesoappendix. pT4a- involvement of the visceral peritoneum (with acellular mucin or mucinous epithelium) pT4b- direct involvement of adjacent organs or structures.
	Peritoneal implants.	EIVA	M1a	Intraperitoneal acellular mucin without neoplastic epithelium in the disseminated peritoneal mucinous deposits.
			M1bG1	Intraperitoneal dissemination containing tumor cells with low-grade cytologic atypia without SRC. Low cellularity (<20%). No infiltrative invasion of the peritoneum; may be involved with pushing front without desmoplastic reaction. Perineural or lymphovascular invasion rarely observed.
		EIVB	M1bG2	Intraperitoneal dissemination containing tumor cells with mixture of low- and high-grade cytologic atypia without SRC. High cellularity (>20%). Infiltrative invasion of the peritoneum and into adjacent organs. Perineural or lymphovascular invasion may be present.
			M1bG3	Intraperitoneal dissemination with tumor cells displaying adverse histological features. High cellularity (>20%). Infiltrative invasion of the peritoneum, adjacent organs. Perineural or lymphovascular invasion may be present.

## 3. Systematic Review on Mucinous Tumors of the Appendix with Peritoneal Dissemination [93]

Our study group conducted a systematic review to gain sufficient historical background to understand the development of current classification systems and the basic histopathological features which define each subcategory.

The PRISMA guidelines were implemented to carry out the systematic review, as shown in Figure 1. MEDLINE and the Cochrane Library were consulted for publications that assessed survival results across the different pathological categories in patients with mucinous neoplasm of the appendix with PD treated with CRS + HIPEC.

The following criteria had to be met for a study to be considered for inclusion:Target population: patients with PD from a mucinous tumor of the appendix treated with CRS + HIPEC.The studies had to report OS and DFS results based on any pathologic classification. In addition, the results had to be shown as median and/or 5-year OS or DFS rate for each histologic category of peritoneal implants. At the least, the survival results of two different histological categories had to be compared in univariable or multivariable analysis.

### 3.1. Results

A total of thirty-eight studies were included. Ronnett’s classification was the most popular as nine studies utilized it. The PSOGI classification was used in six studies and the AJCC 8th edition was used in seven studies.

In the systematic review, nine studies supported a two-tiered classification system for peritoneal dissemination, 12 studies supported a three-tiered system, and two supported a four-tiered classification system.

Since the pioneer publication by Ronnet and Sugarbaker in 1995 to the present day, the classification of mucinous neoplasms of the appendix has been the subject of extensive research. The AJCC 8th edition has collected the descriptions made at PSOGI consensus on both the primary and peritoneal lesions. However, the AJCC 8th edition recognizes two prognostic groups in cases of peritoneal disease: stage IVA includes patients with M1a (acellular mucin) and M1b G1 (low-grade atypia); whereas stage IVB groups together patients with M1b G2 (high-grade atypia) and G3 (high-grade atypia with any component of SRCs). Our study group endorses the division of patients with peritoneal disease into four tiers as proposed by the PSOGI classification, based on the particularly good prognosis of patients with AM in opposition to the particularly poor prognosis of patients with SRC.

In regard to Acellular mucin, several initial studies [76,83] have suggested that patients with acellular mucin disease have a much lower risk of disease recurrence and improved OS compared to those with a low-grade cellular disease. In AJCC terminology, M1a disease seemed to have a lower risk of recurrence than M1bG1 [94]. The clarity gained in pathological reporting as a result of the PSOGI consensus allowed further study groups to assess the uniqueness of patients with acellular mucin. Reghunathan and colleagues [95] observed only one recurrence out of 33 patients with acellular mucinous deposits, with 13 remaining disease-free for more than 3 years (HR 9.8; *p* = 0.025). Furthermore, Choudry et al. [96] compared the risk of recurrence in patients with acellular mucin (19 patients) and scant cellularity (30 patients), to those with moderate cellularity, defined by 2–19 percent of epithelial cells (242 patients), and observed that the risk was higher in the latter with a HR of 4.4 (*p* = 0.02).

There are three studies that applied the PSOGI terminology [94,95,96] that demonstrated that acellular mucin was associated with improved DFS compared to LG-MCP; however, a fourth study [97] found no significant differences.

In regard to signet ring cells, their presence has been traditionally associated with poor outcomes. There were four studies [74,86,88,89] found to confirm this in the multivariate analysis of OS of HGMCP compared to HGMCP-SRC in PSOGI terminology, or M1bG2 compared to M1bG3 in AJCC terminology. Ihemelandu and colleagues [88] observed that median OS decreased from 45.4 months to 18.9 months in patients with moderate–high-grade histology compared to those with SRCs (HR of 1.4, *p* < 0.001). Similarly, Munoz-Zuluaga et al. [86] noted a 90 months median OS in patients with HGMCP that dropped to 26.4 months in those with HGMCP with SRC, with a HR of 2.9 (*p* < 0.001). Pathologists are further challenged by the finding of SRC at the implants since they must distinguish between floating SRCs in pools of mucin and tissue-invading SRCs. Sirintrapun [84] described that the presence of tissue-invading SRC implies significantly worse OS and, therefore, should be reported in the pathology report. However, SRCs floating in pools do not impact the prognosis.

### 3.2. Conclusion

The AJCC 8th edition classification [80] has included many of the peculiarities of appendiceal mucinous tumors that were described at the PSOGI consensus, as well as a specific T staging for LAMN primary lesions. However, only two prognostic groups (stage IVA and stage IVB) were recognized for patients with PD.

There are two study groups that have recently evaluated the prognostic impact of the four-tiered PSOGI classification [77]. In 2017, Huang et al. [92] observed significant differences in OS across the four subgroups with a HR of 3.13, *p* < 0.001. The median OS in patients with AM and LGMCP was not reached; in patients with HGMCP, it was 58.2 months and 31.1 months in those with HGMCP-SRCs. However, in 2018, Baratti and colleagues [79] were unable to reproduce similar results and concluded that the two-tiered WHO classification (HR 1.48, 95% CI 1.04–2.10; *p* = 0.028) correlated better with OS than the PSOGI classification (HR 1.22, 95% CI 0.93–1.59; *p* = 0.149). In their discussion, they argued that having more subgroups reduces the number of patients in each, which decreases statistical power.

CRS + HIPEC is the gold standard treatment for mucinous appendiceal tumors with PD [3]. However, its major drawback lies in its high morbidity and mortality rates [98], therefore adequate patient selection is paramount. For this, a universal language that captures the prognostic implications behind specific pathologic features is urgently needed. This will, in turn, aid the development of management protocols for this disease. The existing scientific literature suggests that the best current classification system is the four-tiered PSOGI classification system, however, it has not been established yet as such.

## 4. Which Classification System Defines the Best Prognosis of Mucinous Neoplasms of the Appendix with Peritoneal Dissemination: TNM or PSOGI [99]?

The aim of this study was to evaluate the prognostic impact of PSOGI and AJCC 8th edition classification systems of mucinous appendiceal neoplasms with PD. A retrospective study was carried out using a prospective registry of consecutive patients treated by the Peritoneal Carcinomatosis Unit at our center. Patients treated with CRS + HIPEC for a disseminated appendiceal mucinous neoplasm between January 2009 and December 2019 were included. There were two expert pathologists in peritoneal surface malignancies that reviewed the pathology slides obtained during the CRS + HIPEC procedure and cases were reclassified adhering to the criteria set at the PSOGI consensus [77] and AJCC 8th edition classification. Survival analysis evaluated the impact of each classification system (PSOGI vs. TNM) on OS and DFS while the concordance-index evaluated their predictive power.

### 4.1. Pathological Evaluation

A microscopic assessment of the appendix differentiated between benign lesions, LAMN, HAMN, and mucinous adenocarcinoma (MAC). Peritoneal implants were classified into AM, LGMCP, HGMCP, and HGMCP-SRC. AM was defined by the absence of epithelial cells and a granulation-like response of the peritoneum. Mucinous deposits where epithelial cells showed low-grade atypia and constituted <20% of the tumor volume were graded as LGMCP, whereas HGMCP when the cellular component was more abundant (>20%) and displayed signs of high-grade atypia. At least >10% SRC component was required for a patient to be classified as HGMCP-SRC. Careful examination was taken to discriminate SRCs from degenerative SRC-like tumor cells, and their disposition was recorded (floating within mucin deposits or invading tissue). Patients were also staged according to the AJCC 8th edition criteria as stages IVA or IVB [80,100]. Essentially, AM and LGMCP peritoneal implants were graded as stage IVA, while HGMCP and HGMCP-SRC implants were graded as stage IVB.

### 4.2. Results

The primary appendix tumor was available for histological examination in 66 cases. Of these, 21 cases (31.8%) were classified as LAMN; 36 (54.5%) as MAC; 5 (7.6%) as MAC with SRC; and 4 (6.1%) as signet ring cell carcinomas (SRCC). No cases matching the criteria for HAMN were observed. The examination of the peritoneal implants reclassified patients as AM in 20 cases (21.1%), LGMCP in 53 (55.8%), HGMCP in 8 (8.4%), and HGMCP-SRC in 14 (14.7%). There was a significant correlation between the grade of the primary appendix tumor and the grade of the peritoneal disease (*p* < 0.001). There were seven LAMN cases that developed AM implants (33.3%) and 14 developed LGMCP (66.7%); there were seven MAC (six well-differentiated and one moderately differentiated) presented AM peritoneal lesions (19.4%), 23 (17 well-differentiated and six moderately differentiated) presented LGMCP (63.9%), five moderately differentiated developed HGMCP (13, 9%) and one poorly differentiated presented HGMCP-SRC (2.8%). All MACs with SRC or SRCC developed with HGMCP-SRC. The 30-day perioperative mortality rate was 2.1%. There was one death on the eighth postoperative day due to massive pulmonary thromboembolism; the second occurred on the thirteenth postoperative day due to mitral endocarditis and septic shock. The Clavien–Dindo grade III/IV perioperative complication rate was 41.4%. The histological subtype was not associated with postoperative morbidity and mortality.

The series median follow-up was 49.2 months. There were six patients lost to follow-up and 11 out of 89 patients died. The median OS of the entire cohort was not achieved and 86.1% of patients were alive at 5 years. Disease relapse was detected in 39 out of 89 patients. The median DFS was 64.7 months (46.1–83.4) and 5-year and 10-year DFS rates were 51.5% and 43.8%, respectively.

Factors that were significantly associated with survival on the univariate analysis were preoperative and postoperative SCT, PCI, CC score, LN status, and PSOGI and AJCC classifications. The survival results for each of the PSOGI and AJCC 8th edition subgroups are shown in Figure 2. The median OS was not reached in AM or LGMCP and was 41.4 months in HGMCP and 56.3 months in HGMCP-SRC (*p* = 0.002). Pairwise comparisons found significant differences when comparing AM and HGMCP-SRC (*p* = 0.006) and LGMCP and HGMCP (*p* = 0.001). Similarly, the median OS was not reached in stage IVA patients and was 56.3 months in stage IVB (*p* < 0.001).

Once adjusted to other possible confounding factors, both classification systems PSOGI and AJCC were significantly correlated with OS with similar HR; 10.2 (*p* = 0.039) and 7.7 (*p* = 0.002), respectively, and they had similar discriminative capacities (c-index values of 0.685 and 0.669, respectively). Out of the previously mentioned factors, only PCI > 21 retained its significance in the multivariate analysis with a HR of 11.4, *p* = 0.022.

Factors that were found to be significantly associated with DFS on the univariate analysis were preoperative SCT, PCI, CC score, preoperative elevated TM, LN status, and PSOGI and AJCC classifications. The DFS results for each of the PSOGI and AJCC 8th edition subgroups are shown in Figure 3. Once again, median DFS was not reached in patients with AM but it was 60.9 months in LGMCP, 13.6 months in HGMCP, and 8.8 months in HGMCP-SRC (*p* < 0.001). A pairwise comparison method found significant differences in the comparison of AM with LGMCP (*p* = 0.029), HGMCP (*p* < 0.001), and HGMCP-SRC (*p* = 0.002); and LGMCP with HGMCP (*p* = 0.008) and HGMCP-SRC (*p* = 0.013). In the AJCC classification, the median DFS of stage IVA patients was not reached and was 9.1 months in stage IVB patients (*p* < 0.001). Both classification systems were significantly associated with DFS in the multivariate analysis with a respective HR of 12.7 (*p* = 0.001) and 3.7 (*p* < 0.001). AM was found to have significantly lower recurrence rates than every other histological subgroup (AM vs. LGMCP (HR 4.95, *p* = 0.03); AM vs. HGMCP (HR 17, *p* = 0.001) and AM vs. HGMCP-SRC (HR 12.7, *p* = 0.001). However, analysis of the concordance index suggested that both classifications had similar discriminative power (0.669 and 0.623 for PSOGI and AJCC, respectively). The other risk factors associated with lower DFS in the multivariate analysis were postoperative SCT, PCI > 21, and elevated TM.

### 4.3. Discussion

The clinical outcome of patients with mucinous neoplasms of the appendix with PD treated with optimal CRS + HIPEC can vary greatly. Histology has an important role in determining the clinical course of the disease [68,69,70,72,101,102]. The greatest achievement of recent classification systems [77,80,100] has been to provide concrete pathological criteria to define the different histological grades in an attempt to universalize the terminology used and standardize treatment protocols.

The PSOGI classification system [77] supports the existence of four prognostic subgroups, whereas AJCC 8th edition [80,100] recognizes two tiers. The findings from our series appear to agree with the two prognostic groups outlined by the AJCC 8th edition [80,100]. The 5-year OS rates of AM and LGMCP (95% and 94%, respectively) were excellent and comparable to those of stage IVA (94.3%). Similarly, HGMCP and HGMCP-SRC subgroups had considerably worse prognoses with 5-year OS rates of 50% and 30% (*p* = 0.434) comparable to stage IVB (39.7%). However, the following points must be highlighted:

Firstly, the negative impact of the presence of SRC on survival could not be demonstrated in our series. However, this has been repeatedly observed by many study groups resulting in the criticism of the AJCC 7th edition [73] as patients with and without SRC were grouped into the same prognostic group [75]. Ihemelandu and Sugarbaker [88] and Munoz-Zuluaga et al. [86] noticed significantly lower median OS in patients with SRC compared with those without SRC: 18.9 (HR 1.4, *p* < 0.001) and 26.4 (HR 2.9, *p* < 0.001) months, respectively. The reduced number of patients with SRC in our series (*n* = 14) as well as the unexpectedly high survival outcomes achieved (median OS 56.3 months) may explain why our series could not verify the negative impact of SRC on survival. Secondly, when recurrence is considered, the study we conducted emphasizes the higher recurrence risk of high-grade tumors, as previously noted by many other study groups, [94,95,96] as well as the exceptionally low propensity of AM to recur when compared with cellular counterparts. This was clearly noted in our series as the 5-year DFS rates of patients with AM were much higher than those with LGMCP (82.2% vs. 51.2% with a HR of 4.95, *p* = 0.03). Although the AJCC 8th edition included a distinct category for this particularity (M1a), patients were later grouped together with LGMCP (M1b G1) into stage IVA, therefore, the lower risk of recurrence of AM disappears in the AJCC 8th edition.

Histology is currently still considered to be the main factor driving patient prognosis, which, in turn, conditions treatment and follow-up schemes. For example, the use of SCT which was previously administered more frequently as this disease was originally thought to be incurable, has been found to be of benefit in high-grade histologies [58]. This has been adapted by recent clinical guidelines [3,4] that contemplate the use of neoadjuvant SCT to reduce tumor burden in high-grade tumors where optimal CRS (CC0-1) is not initially feasible but not in low-grade cases. The Peritoneal Malignancy Institute (PMI) Basingstoke has assessed the influence of histology on follow-up protocols recommending different surveillance schemes based on a patient’s final histology [103]. Similarly, the Manchester study group [104] advocated for a limited surveillance scheme of annual CT scans for up to 5 years in patients with AM arising from a primary LAMN after observing a 3% recurrence rate (at 12 and 56 months). In general, we agree with this viewpoint, as long-term follow-up protocols may be ineffective in AM and may result in unnecessary radiation exposure.

However, one key finding that we would like to emphasize from the results of our study is that histology on its own has limited discriminative power as none of the classification systems scored a c-index value higher than 0.7 predicting OS and DFS. Therefore, other prognostic factors (serological TM [105], biological markers [106,107] …etc.) should be taken into account to achieve better patient stratification. We would also like to highlight that some histological landmarks still have to be properly defined such as the number of slides that should be assessed to confidently categorize a patient as AM [108] and the clear pathologic description of malignant SRC.

### 4.4. Conclusions

The AJCC 8th edition and the PSOGI classification systems have demonstrated a similar capacity of stratifying patients into prognostic groups in our patient cohort. When considering DFS, the PSOGI classification seems to provide a slightly better prognostic stratification. However, histology’s discriminative capacity is insufficient on its own. Other prognostic indicators must be identified in order to improve patient classification and establish more efficient treatment and follow-up regimes.

## Figures and Tables

**Figure 1 cancers-15-03426-f001:**
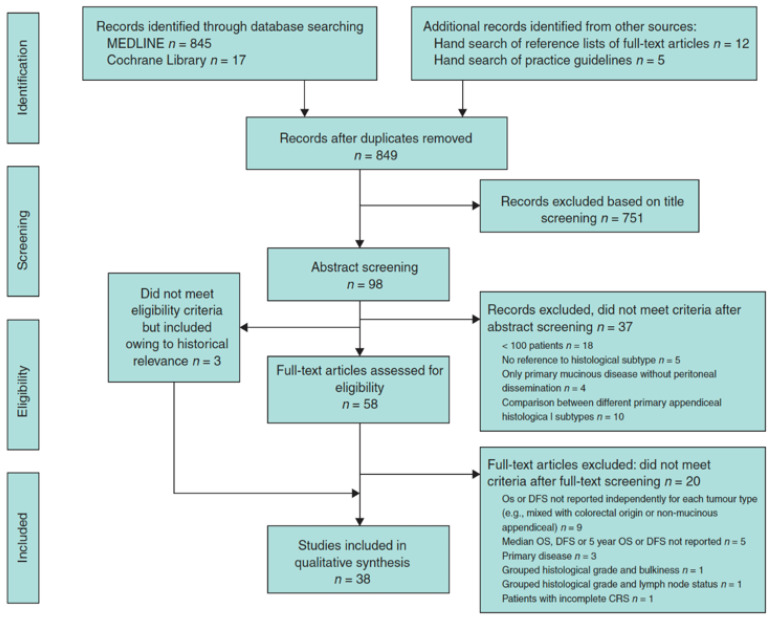
Flow chart showing selection of studies for review.

**Figure 2 cancers-15-03426-f002:**
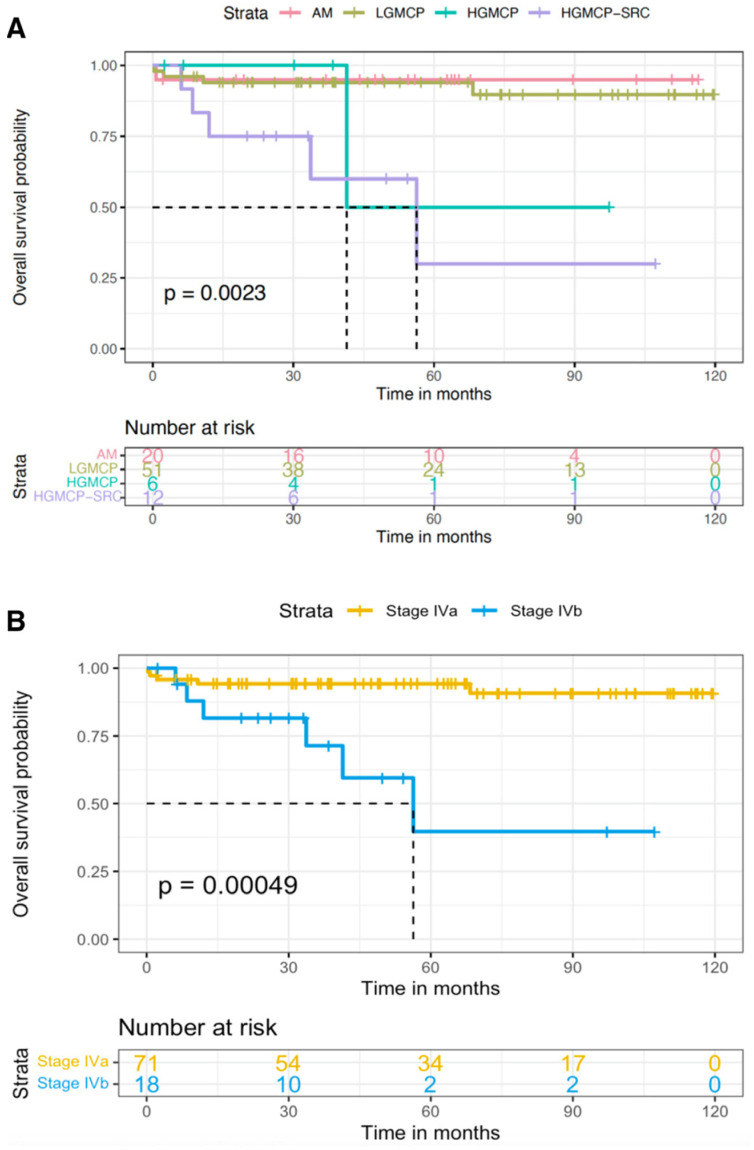
Overall survival according to Peritoneal Surface Oncology Group International classification of peritoneal implants (**A**) and eighth edition of the American Joint Committee on Cancer (**B**) [99]. AM, acellular mucin; HGMCP, high-grade mucinous carcinoma peritonei; HGMCP-SRC, HGMCP with signet ring cells; LGMCP, low-grade MCP.

**Figure 3 cancers-15-03426-f003:**
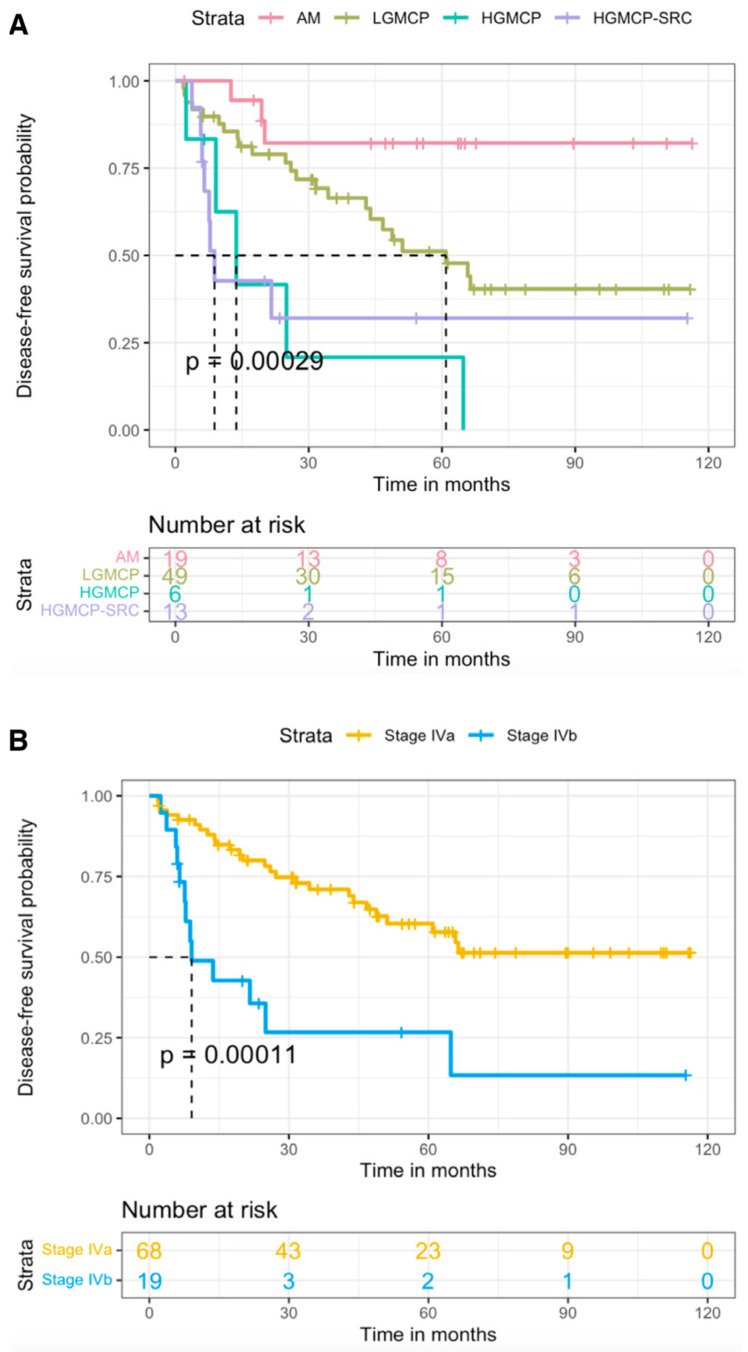
Disease-free survival according to Peritoneal Surface Oncology Group International classification of peritoneal implants (**A**) and eighth edition of the American Joint Committee on Cancer (**B**) [99]. AM, acellular mucin; HGMCP, high-grade mucinous carcinoma peritonei; HGMCP-SRC, HGMCP with signet ring cells; LGMCP, low-grade MCP.

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
