# Peer review of "Appendiceal Mucinous Neoplasms: From Clinic to Pathology and Prognosis"

_cancers, 2023, doi:10.3390/cancers15133426_

Round 1

Reviewer 1 Report

Good work has been done.

Comparison between PSOGI and JCCC histology concerning mucous appendiceal neoplasms is correct and well conducted. It is a significant paper in order to confirm histology classification for these tumours. It adds a confirmation of the most relevant MAN classification, PSOGI Adoption of the PSOGI seems to more suitable for pathologists Methodology is correct. Conclusion is consistent and references appropriated.
This paper is a comparison of classifications method. It is not a new method so it cannot be considered pivotal but it is well conducted

No major corrections are necessary unless of the Summary which test is not of Scientific style.

Author Response

Thanks for the feedback

Reviewer 2 Report

This manuscript reviews the classification guidelines for appendiceal mucinous neoplasms, focusing on PMP. 

General concerns:

-       The review structure is confusing. The manuscript should be reorganized. Section 1 and 2 may be combined and much condensed, with sub-sections discussing "clinical aspects," "tumor markers," and so on.

-       The manuscript is not well written. English language has to be improved. There are errors in the data provided throughout the paper (for example, N=720 on line 397 should be N=750). There are several redundant paragraphs that need be rewritten. There are errors in the section numbering and abbreviations that are not adequately defined. 

-       There are important references that should be included such as: DOI: 10.1186/s12885-023-10545-7, DOI: 10.1245/s10434-021-10372-9, DOI: 10.1016/j.ejso.2023.03.206. 

-       Sentences 306–309 should be revised because the statement is not properly supported by the literature. 

-       DOI: 10.1245/s10434-021-10372-9 and DOI: 10.1016/j.ejso.2023.03.206 should be included in Table 1.

-       The methods used to conduct the systematic review should be better described. A table containing all of the selected papers as well as the information derived from them must also be included in the manuscript. A risk of bias assessment should also be included in the manuscript.

-       Section 4 of the manuscript should be removed because is an extensive description of the paper “Why classification system defines best prognosis of mucinous neoplasms of the appendix with peritoneal dissemination: TNM vs PSOGI?” already published by the authors. However, a summary of the information can be included in the rearranged section 1. 

Extensive editing of English language required. 

Author Response

Thanks for the feedback

Section 1 INTRODUCTION makes an approach to the subject and describes the content of the article with the conclusions reached, as indicated by MDPI template.

Section 1 and 2 has been review for English edition and for mistakes. They have been newly written 

Sentences 306–309 have been rewritten and properly supported by the reference 46 Yonemura ….

 line 397: N=720 has been changed to N=750 and rewritten

Table 1 is already published; therefore it is not possible to include new references.

The systematic review is already published. Please, review original article for complete information.

Section 3 and Section 4 has been reviewed for English editing. This sections are a summary of two articles published by our group (BJS Open, 2021, zrab059, DOI: 10.1093/bjsopen/zrab059 y  J Clin Path DOI:10.1136/jclinpath-2021-207883) and do not include more than a brief summary of the results and discussion so as not to make this review too long.

Reviewer 3 Report

In my opinion, the analyzed topic is interesting enough to attract the readers’ attention. This study examined all the aspects of appendiceal mucinous neoplasms. I think that the abstract of this article should be reorganized in order to highlight the aim of the study. The discussion could be studied in depth and extended. Maybe, the assessment of frailty in these kind of patients could be useful and should be analyzed. In particular, I suggest this article to get deeper in the topic: The role of preoperative frailty assessment in patients affected by gynecological cancer: a narrative review Ottavia D’Oria, Tullio Golia D’Auge, Ermelinda Baiocco, Cristina Vincenzoni, Emanuela Mancini, Valentina Bruno, Benito Chiofalo, Rosanna Mancari, Riccardo Vizza, Giuseppe Cutillo, Andrea Giannini Vol. 34 (No. 2) 2022 June, 76-83 doi: 10.36129/jog.2022.34 . Because of these reasons, the article should be revised and completed. Considered all these points, I think it could be of interest for the readers and, in my opinion, it deserves the priority to be published after revisions.

A moderate review of English language should be performed

Author Response

Thanks for the feedback

We have remade the abstract

We have included evaluation of frailty in 2.6 CRS + HIPEC and a reference (J Am Coll Surg 2018;226:173e182. On behalf of the American College of Surgeons). We do not have included the reference suggested because is specific of Gynecologic malignancies.

Round 2

Reviewer 2 Report

The manuscript has been reviewed and improved. However, there are some minor changes that should be made:

-   English editing still needed.

-   Lines 124-125: DFS and OS should be defined.

-   Line 215: 94% is wrong. The correct number is 96.4%

-   Section 2.9 is 2.10.

Although English has clearly improved, editing is still required.

Author Response

Thank you for the feedback

English editing has been reviewed

  •    Lines 124-125: DFS and OS should be defined. DFS and OS have been defined

-   Line 215: 94% is wrong. The correct number is 96.4%. Number has been corrected

-  Section 2.9 is 2.10. Section number has been changed

Reviewer 3 Report

I read with great interest the Manuscript titled " Appendiceal Mucinous Neoplasms: from clinic to pathology and prognosis”, topic interesting enough to attract readers' attention. 

The quality of the manuscript has improved thanks to the changes made. The methodology is accurate, and results are supported by data analysis. The discussions are clear and comprehensive. References are relevant to the search

Considering all these points, I think it could be of interest to the readers and, in my opinion, it deserves the priority to be published.

Author Response

Thank you for the feed back

A new round of English edition has been done

-   Lines 124-125: DFS and OS should be defined. DFS and OS have been defined

-   Line 215: 94% is wrong. The correct number is 96.4%. Number has been corrected

-  Section 2.9 is 2.10. Section number has been changed